# Adolescents’ Physical Activity and Depressive Symptoms: A Psychosocial Mechanism

**DOI:** 10.3390/ijerph19031276

**Published:** 2022-01-24

**Authors:** Liang Shen, Xiangli Gu, Tao Zhang, Joonyoung Lee

**Affiliations:** 1School of Physical Education, Shanghai University, Shanghai 200444, China; 2Department of Kinesiology, University of Texas at Arlington, Arlington, TX 76019, USA; xiangli.gu@uta.edu; 3Department of Kinesiology, Health Promotion and Recreation, University of North Texas, Denton, TX 76203, USA; tao.zhang@unt.edu; 4Department of Health, Physical Education, and Recreation, Jackson State University, Jackson, MS 39217, USA

**Keywords:** adolescents, physical activity, depression, theory of planned behavior

## Abstract

Guided by the Theory of Planned Behavior (TPB; Ajzen, 1991), this study aimed to test a hypothesized path model of TPB variables (i.e., attitude toward behavior, subjective norm, perceived behavioral control, and intention) with physical activity (PA) and depressive symptoms. A total of 792 Chinese adolescents (*M_age_* = 16.9; *SD* = 0.82; 54.5% females) completed previously validated questionnaires assessing their TPB variables, PA, and depressive symptoms. Correlation analysis revealed TPB variables were positively associated with PA (*p* < 0.01), and negatively correlated to depressive symptoms (*p* < 0.01). The path analyses indicated that the hypothesized model produces a goodness of fit (χ^2^/df = 16.14/5, CFI = 0.99, RMSEA = 0.06). The findings support the theoretical tenets of TPB and provide empirical evidence of the psychosocial mechanism of PA and depressive symptoms in Chinese adolescents. It suggests that building PA intervention strategies while considering the TPB framework may promote adolescents’ physical and mental health.

## 1. Introduction

The trends of increasing major depressive symptoms across all ages (i.e., children to elders) have centered the prevention efforts for depression [1,2,3,4,5,6,7]. Of particular note is the steep rise in major depressive episodes among adolescents (14–20% between 15- and 18-year-olds; [8,9,10]). Significant data regarding Chinese adolescents also showed the notably higher prevalence of depressive symptoms after 2016 compared to before 2000, from 18.4% to 26.3% [11]. Research evidence indicates that adolescent depression can lead to self-harm, suicidality, substance/drug misuse, poor academic achievements, and impaired social functioning [1,12]. Given that the early onset of depressive episodes contributes to a significant risk of chronic mental illness [13,14], proactive efforts in adolescence are needed to reduce and prevent severe major depressive episodes [1,15].

The significant health benefit and protective effect of regularly engaging in physical activity (PA) is well-documented, including improved physical fitness [16,17,18,19], self-concept/worth [20,21], cognitive functioning [22], healthy dietary habits [17,23], and well-being [21,24,25]. Importantly, accumulative evidence has shown an inverse relationship between PA and depression, indicating that adolescents who engaged in higher moderate-to-vigorous PA (MVPA) tended to have fewer depressive symptoms [3,26]. Systematic review and meta-analysis studies also suggested that increasing PA can be used as a therapeutic intervention plan for the prevention and alleviation of depression among adolescents [1,27,28,29]. Encouraging adolescents to engage in daily PA would play an important role in preventing and reducing early-onset depressive episodes [30,31]. However, a recent population-based survey study from 146 countries reported that more than 77% of adolescents aged 11–17 years engaged in insufficient PA regardless of gender [32]. Specifically, less than one-third of Chinese adolescents living in urban and rural areas achieve the recommended daily 60-min MVPA [33]. Furthermore, adolescence represents a period of rapid development and significantly decreased PA participation in this age group was noticed [34,35,36,37]. Increasing numbers of experimental studies also support that increased PA participation may result in a small but positive effect on depression among youth [25,38,39]. Hughes and colleagues [40], for example, found that depressed adolescents (12–18 years) demonstrated a rapid reduction of depression after the 6–8 weeks of PA intervention, but no significant differences were observed after a 12-week intervention. To date, the underlying mechanism of the positive effect of PA on depression has not been investigated.

Researchers have applied various models over the past few decades to acquire better knowledge about adolescents’ PA behaviors and mental health. The theory of planned behavior (TPB; [41]) is one of the most widely-used and predictive models to comprehend individuals’ PA intention and behaviors [42,43]. TPB was originally developed and extended by Ajzen and Fishbein’s theory of reasoned action [44], which involved three main constructs influencing individuals’ behaviors: (a) attitude toward behavior, (b) subjective norm, and (c) behavioral intention. The TPB indicates that individuals are more likely to intend to execute behaviors (i.e., behavioral intention) if they positively evaluate the outcomes of behavior (i.e., attitude toward behavior) and believe that significant people in their lives want them to perform the behavior (i.e., subjective norm). It was also noted that the *perceived behavioral control* construct of the TPB model is a direct predictor of individuals’ intention and behaviors as individuals’ intentions of and decisions surrounding performing behaviors may be related to perceptions about whether they believe they can do them or not [41].

The TPB has been proven to be a successful theoretical model that shows that attitude toward behavior, subjective norm, and perceived behavioral control can shape individuals’ PA intention and PA [43,45,46,47]. To what extent this TPB-based psychosocial mechanism may be utilized to understand depressive symptoms in adolescents remains unclear. Although previous studies demonstrated the bidirectional relationship between TPB variables and PA [45,47,48], as far as we know, the TPB has not yet been directly applied to depression in an adolescent’s sample. Guided by the TPB model, this study attempted to test a hypothesized path model between TPB variables (i.e., attitude toward behavior, subjective norm, perceived behavioral control, and PA intention), PA, and depressive symptoms among Chinese adolescents (see Figure 1). It was hypothesized that the TPB variables and PA intention would significantly predict adolescents’ PA behavior and depressive symptoms, respectively.

## 2. Materials and Methods

### 2.1. Participants and Procedures

A total of 860 adolescents were recruited from four public high schools in Shanghai, China, and voluntarily participated in this study. Due to missing or incomplete data (*n* = 68), the final sample with 792 Chinese adolescents (*M*_age_ = 16.9; *SD* = 0.82; 54.5% females) was included in the data analysis. The participating schools were all located in the same district and had similar PA environments (e.g., physical education [PE] curriculums, equipment, facilities, size of PA spaces).

The cross-sectional design was used in this study. The university institutional review board (IRB) reviewed and approved the research protocol (SHU-2017-00145). Before collecting data, we obtained parental consent and student assent from participating school principals and PE teachers. Participants completed validated questionnaires in classrooms during the PE classes. Our research assistants monitored and guided the data collection procedure at each site to ensure data quality.

### 2.2. Measures

#### 2.2.1. TPB Variables

Students’ TPB variables were measured using previously validated questionnaires among adolescents in Motl et al.’s study [49]. The questionnaires included each construct of TPB variables related to PA: attitude toward behavior (8 items), subjective norm (6 items), perceived behavioral control (4 items), and PA intention (4 items) rated on a 5-point Likert scale with responses ranging from 1 (“Disagree a lot,” “Very bad,” or “Very difficult”) to 5 (“Agree a lot,” “Very good,” or “Very easy”) depending on the questions. The sample items were “If I were to be physically active during my free time on most days, it would be fun” in the attitude toward behavior, “My best friend thinks I should be physically active during my free time on most days” in the subjective norm, “If I want to be I can be physically active during my free time on most days” in the perceived behavioral control, and “I intend to be physically active during my free time on most days” in the PA intention [49,50]. Motl et al.’s TPB variable questionnaires [49,50] have been highly validated and reliable among adolescents in previous studies [51,52] and showed acceptable reliability of all TPB variables (Cronbach’s *α* ranging from 0.90 to 0.93) in our sample.

#### 2.2.2. Physical Activity (PA)

Participants’ PA was measured using the PA questionnaire for adolescents (PAQ-A; [53]). The questions ask about the general MVPA levels in spare time activity, PE, lunch, right after school, evenings, and weekends of a 7-day period. The PAQ-A involves six items on a 5-point Likert scale (1 = low PA; 5 = high PA), and higher scores represent higher levels of PA. The mean score of all responses was used as the PA summary score. We also used the recommended cutoff score (*M* > 2.75) to identify the proportion of active vs. inactive adolescents in PAQ-A [54]. This questionnaire has shown acceptable reliability and validity in previous research among adolescents [55,56] and indicated sufficient reliability in our sample (Cronbach’s *α* = 0.78).

#### 2.2.3. Depression

The short form (10 items) of the Center for Epidemiological Studies-Depression scale (CES-D; [57]) was used to measure participants’ depressive symptoms. The 10 items ask students to rate how often in the previous week they experienced depressive symptoms, such as depressed mood, feeling lonely/hopelessness, and sleep disturbance. The questionnaire includes a 4-point Likert scale, ranging from 0 (“Rarely or none of the time–less than 1 day”) to 3 (“Most of all the time–5 to 7 days”). Two items were reverse coded, and the sum scores of all items were used in the final analysis. Possible scores range from 0 to 30, and a score greater than ten indicates significant depressive symptoms [57]. The CES-D has been shown to be reliable and valid for measuring depressive symptoms among adolescents [58,59], and demonstrated a good reliability (Cronbach’s α = 0.80) in our sample.

### 2.3. Data Analysis

Descriptive statistics were analyzed to show summarized data (mean, standard deviation [SD], and normality) using the statistical package for social sciences (SPSS 28.0) software. The skewness and kurtosis of all variables were examined to check if they are normally distributed (between −2 and +2) [60]. The Pearson correlation coefficient was used to examine the relationships between TPB variables (attitude, subjective norm, perceived behavioral control, and PA intention), PA, and depression. The hypothesized path model was tested using the structural equation modeling techniques (AMOS 26.0, IBM Corp, Armonk, NY, USA; [61]). The goodness of fit of the model was evaluated using the following index standards [60,62]: chi-square (χ^2^), normed fit index (NFI), incremental fit index (IFI), comparative fit index (CFI) greater than 0.95 (0.90 is acceptable), and root-mean-square error of approximation (RMSEA) less than 0.06 (0.08 is acceptable).

## 3. Results

Skewness (which ranged from −0.51 to 0.59) and kurtosis (which ranged from −0.16 to 0.41) indicated normal distribution of the study variables. The means and standard deviations (SDs) of TBP variables are described in Table 1, showing attitude toward behavior (*M* = 3.86, *SD* = 0.75), social norm (*M* = 3.67, *SD* = 0.75), perceived behavioral control (*M* = 3.45, *SD* = 0.96), and PA intention (*M* = 3.77, *SD* = 0.87). Participants had a lower PA level (*M* = 2.26, *SD* = 0.78) compared to the average PA norm score in PAQ-A (*M* = 2.75; [54]). Adolescents in this study reported significant depressive symptoms (*M* = 10.80, *SD* = 5.64; [57]). These findings also indicated that 76% of the participants are inactive (PAQ-A < *M* = 2.75; [54]), and 55% of the sample showed clinically significant depressive symptoms (CES-D > = 10; [57]). Furthermore, the correlation results indicated significantly positive associations among all TPB variables (*r* ranged from 0.52 to 0.65, *p* < 0.01). Most of the TPB variables were positively corelated with PA (*rs* ranged from 0.26 to 0.48, *p* < 0.01), whereas they were negatively associated with depression (*rs* ranged from −0.19 to −0.12, *p* < 0.01). No significant association between the social norm and depressive symptoms was found (*p* > 0.05).

The final hypothesized path model produces a good fit to the data: χ^2^/df = 16.14/5, *p* < 0.01, NFI = 0.99, IFI = 0.99, CFI = 0.99, RMSEA = 0.06. The model accounted for 24% and 3% of the variance toward PA and depression, respectively. Path coefficients suggested that attitude (*β* = 0.38, *p* < 0.01), subjective norm (*β* = 0.16, *p* < 0.01), and perceived behavioral control (*β* = 0.30, *p* < 0.01) were positively associated with PA intention, only perceived behavioral control emerged as a significant direct predictor of PA behavior (*β* = 0.41; *p* < 0.01). PA intention directly predicted PA behavior (*β* = 0.12, *p* < 0.01) and depression (*β* = −0.13; *p* < 0.01), respectively (see Figure 2).

## 4. Discussion

Although significant issues regarding physical inactivity and high rates of depressive symptoms in adolescents are rising, there is still limited evidence establishing a psychosocial mechanism between PA and depression in adolescence. This might be because the prevalence of childhood and adolescent depressive symptoms tended to be underestimated compared to the prevalence of adult symptoms [63,64]. The bidirectional relation between PA and depression has been well-documented; however, an increasing number of experimental studies reporting that increased PA has small but positive effects on mental health [38]. Lubans and colleagues [38] mapped three potential mechanisms (i.e., behavioral, psychosocial, and neurobiological) toward the relation between PA and menta health outcomes in their systematic review. According to the psychological mechanism proposed in Lubans et al.’s conceptual model, adolescents experience a period of rapid development influenced by various psychosocial attributes (i.e., attitude, self-concept). Given the fact that Chinese adolescents showed the notably higher prevalence of depressive symptom, this study aimed to test a hypothesized path model among Chinses adolescents guided by TPB. The findings support the theoretical tenets of TPB and provide empirical evidence of the moderate associations of key TPB variables (i.e., attitude toward behavior, perceived behavioral control) and PA with depressive symptoms. The results also demonstrated that attitude toward behavior, subjective norm, and perceived behavioral control domains in TPB construct indirectly affected the students’ depressive symptoms through PA intention and behavior. The findings are in line with previous research on the direct associations between TPB variables and PA [43,45,48].

The prevalence of depressive symptoms among adolescents has been widely acknowledged [8,10], especially considering the rising trends of depression in Chinese adolescents [11]. In this adolescents sample, more than half of participants showed depressive symptoms (55% participants; [57]). This study also revealed that more than 70% of Chinese adolescents participated in less than recommended 60 min/per day of MVPA (*M* = 2.75; [54]). These descriptive results indicate that it would be imperative to understand adolescents’ social and cognitive predictors of PA behavior. Consistent with the TPB model, the results suggest behavioral intentions are posited as direct and significant predictors of PA intention and behavior which may influence the subsequent behavioral enactment. Attitudes, subjective norms, and perceived behavioral control are proposed to influence behavioral intent, a proximal predictor to the enactment of health-related behavior. Aligned with the findings from a meta-analyses [65], it was suggested that attitudes, subjective norms, and perceived behavioral control predict PA intentions (accounting for 53% of the variance) and intentions emerged as a direct predictor of PA behaviors (accounting for 24% of the variance).

The unique contribution of this study was to apply a holistic approach demonstrating the specific pathway of TPB variables and PA with adolescents’ depressive symptoms. It provides the preliminary evidence that the TPB model can be applied to understand PA behavior, but also for mental health outcomes, such as depression. To date, this is one of the first studies to examine the TPB-based psychosocial mechanism toward depression through PA intention and behavior. The TPB has been utilized in the literature to link the problem of seeking mental health services among young adults [66,67]. For example, a recent study [67] reported that positive attitudes toward behavior was a consistent predictor in help-seeking intention for mental health services (i.e., depression). The established psychosocial pathway between TPB construct, PA and depression in this study provide empirical insights on how and under what conditions depressive symptoms reduction occur, which would facilitate the design and implementation of the future mental health intervention for high school adolescents in China. Results also suggest that a PA program/intervention that influences positive attitudes and enhances perceived self-efficacy and knowledge may increase PA intentions, which may reduce the likelihood of elevated depressive symptoms during adolescence. For example, a longitudinal study [68] found that adolescents girls who had high initial levels of self-efficacy had the lowest decline in PA participation and are less likely to increase depressive symptoms over time compared to their counterparts. These results may explain the observed depression prevention benefits of PA intervention programs reported in the current literature [69].

Although a large sample size examining the hypothesized path model in this study was the strength of our study, several limitations need to be acknowledged. Due to the nature of applying a cross-sectional study design, the results may not determine cause-and-effect relationships; thus, an experimental or longitudinal design is suggested to investigate the hypothesized path model among the study variables over time in high school students. This study was solely focused on Chinese adolescents; thus, the generalizability of the findings is limited. Future research is needed to collect data from diverse ethnic groups to enhance the generalizability of the findings by applying a cross-cultural design. Finally, objective PA measurement such as accelerometers or pedometers can be used in the future studies to capture intensity levels of PA participation among this age group.

## 5. Conclusions

In conclusion, the findings suggest that the TPB theoretical framework can play an important role in promoting adolescents’ PA behaviors, which can be applied as a theoretical framework for designing and implementing interventions aimed to prevent depressive symptoms. The current results represent the first demonstration of PA intention/behavior’s mediating effect on the TPB–depression relationship among adolescents. Given the prevalence of depressive symptoms in the adolescent population, knowledge about the psychosocial and behavioral factors of depression can aid secondary school health and mental health specialists in tailoring specific programs and messages on depression prevention. Encouraging PA participation and providing developmentally appropriate PA intervention strategies that consider the TPB model (i.e., self-efficacy, positive attitudes) are recommended for physical activity and mental health promotion among Chinese adolescents. Lastly, physical therapy and mental health services can be considered as school resources to support both the physical and mental health of individuals of any age.

## Figures and Tables

**Figure 1 ijerph-19-01276-f001:**
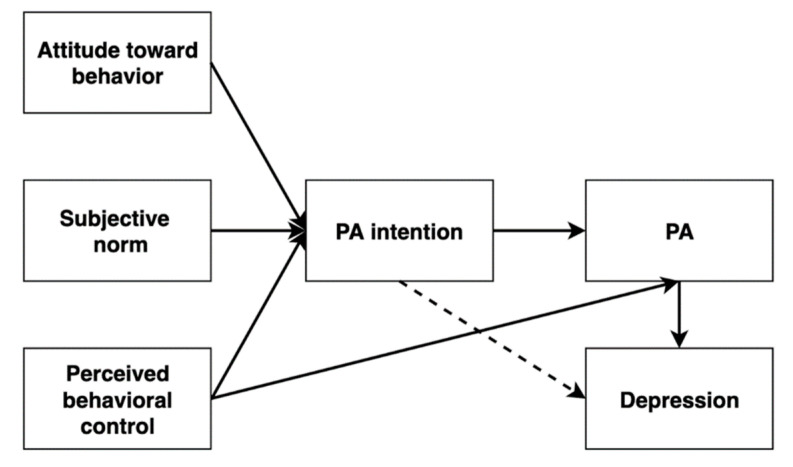
Hypothesized model illustrating relationships among study variables. PA = physical activity.

**Figure 2 ijerph-19-01276-f002:**
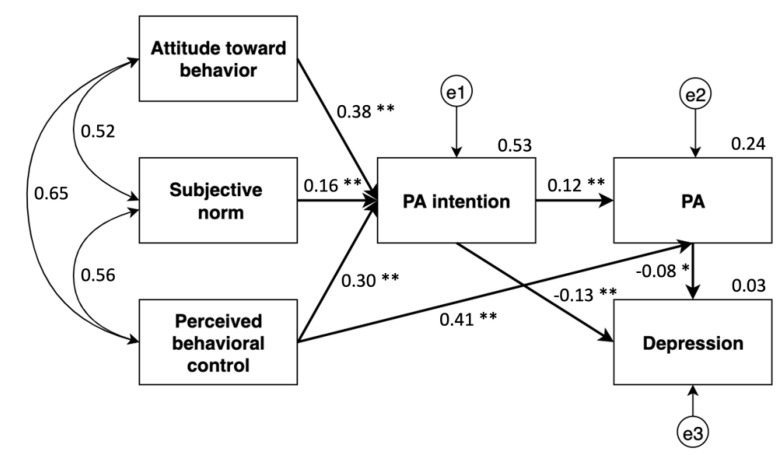
Final structural equation model among study variables; ** *p* < 0.01; * *p* < 0.05.

**Table 1 ijerph-19-01276-t001:** Descriptive statistics and correlations among study variables (*n* = 792).

Variables		*PR*	*M* (*SD*)	1	2	3	4	5	6
1.	Attitude toward behavior	1–5	3.86 (0.75)	-					
2.	Social norm	1–5	3.67 (0.75)	0.52 **	-				
3.	Perceived behavioral control	1–5	3.45 (0.96)	0.65 **	0.56 **	-			
4.	PA intention	1–5	3.77 (0.87)	0.65 **	0.53 **	0.64 **	-		
5.	PA	1–5	2.26 (0.78)	0.37 **	0.25 **	0.48 **	0.37 **	-	
6.	Depression	0–30	10.80 (5.64)	−0.19 **	−0.06	−0.14 **	−0.15 **	−0.12 **	-

*PR* = possible range, *M* = mean, *SD* = standard deviation; ** *p* < 0.01.

## Data Availability

The data presented in this study are available on request from the corresponding author. The data are not publicly available due to consent provided by participants.

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
