# Peer review of "Adolescents’ Physical Activity and Depressive Symptoms: A Psychosocial Mechanism"

_ijerph, 2022, doi:10.3390/ijerph19031276_

Round 1

Reviewer 1 Report

This article focuses on factors improving mental health and physical activity in adolescents.  Given the Covid-19 pandemic, it's likely that depression is found in larger numbers among adolescents.  

Suggestions for editing: this article was very well written and easy to understand.  I suggest deleting a "." after "assent" on line 90 and add "and" before "perceived behavioral control" on line 161.

I encourage the follow-up suggested for a broader population of adolescents including large numbers of individuals from various ethnic groups. 

The Figure 2 presented an excellent depiction of the structural equating model used in this study.

The final sentence in lines 237-240 recommending PT and mental health promotion for adolescents is definitely on target for supporting both physical and mental health for adolescents (and individuals of any age).

Thank you for your clear writing and use of correlations among and across the factors studied.  As a Physical Therapist I commend you for your focus on this age group and the rates of PA and mental health issues.

Reviewer 2 Report

Thank you for the opportunity to revise this manuscript. The topic is interesting and timely. Depression in adolescents is a major risk factor for suicide and it predicts depression and anxiety in adulthood.

The development of depression seems to increase with low levels of physical activity, so it is important to investigate further this aspect.

In general, the study gives an interesting multifactorial impression of the different aspects of Theory of Planned Behavior. However, I have some suggestions to improve the introduction and discussion.

The introduction section should be expanded. It is therefore appropriate to mention studies in which the benefit of promoting physical activity is being experienced.

Good references are:

  • La Torre, G., Mannocci, A., Saulle, R., Sinopoli, A., D'Egidio, V., Sestili, C., ... & Masala, D. (2017). Improving knowledge and behaviors on diet and physical activity in children: results of a pilot randomized field trial. Ann Ig, 29(6), 584-594.
  • Liu M, Wu L, Ming Q. How Does Physical Activity Intervention Improve Self-Esteem and Self-Concept in Children and Adolescents? Evidence from a Meta-Analysis. PLoS One. 2015 Aug 4;10(8):e0134804. doi: 10.1371/journal.pone.0134804. PMID: 26241879; PMCID: PMC4524727.
  • D'Egidio V, Lia L, Sinopoli A, Backhaus I, Mannocci A, Saulle R, Sestili C, Cocchiara R, Di Bella O, Yordanov T, Mazzacane M, La Torre G. Results of the Italian project 'GiochiAMO' to improve nutrition and PA among children. J Public Health (Oxf). 2021 Jun 7;43(2):405-412. doi: 10.1093/pubmed/fdz129. PMID: 31786612.
  • Owen M.B., Curry W.B., Kerner C., Newson L., Fairclough S.J. The effectiveness of school-based physical activity interventions for adolescent girls: A systematic review and meta-analysis. Prev. Med. 2017;105:237–249. doi: 10.1016/j.ypmed.2017.09.018

In the discussion it is necessary to deepen what is meant by limited evidence concerning psycho- social mechanism between PA and depression in adolescence: why aren’t there enough studies on this topic? Isn’t there enough interest? Has the evidence so far been unsatisfactory?

It would be appropriate to describe some experiences in Europe or America about this topic.

Reviewer 3 Report

Thank you for the authors contribution on this study. The manuscript has a good structure and follow guidance of the scientific publication. There is a sort, but brief introduction, however I would suggest to the authors to highlight the chines studies, since this study about chines adolescent. Furthermore, is not clear what county that the authors writing about in line 44 ("However, less than one-third of 43 adolescents (12–19 years old) achieve the recommended daily 60-minutes MVPA”). Please clarify these sentences. Please also add more information on the participants background (e.g., education, family, etc.). One of the big issues of this study that the authors claim they are using SEM, however the SEM using latent variable. In this study there are only observed variables which indicate that the authors using path analysis.
